

# Sixfold fermion near the Fermi level in cubic PtBi$_2$

Setti Thirupathaiah[1, 2], Yevhen Kushnirenko[1], Klaus Koepernik[1, 3],
Boy Roman Piening[1], Bernd Büchner[1, 3], Saicharan Aswartham[1],
Jeroen van den Brink[1, 3], Sergey Borisenko[1] and Ion Cosma Fulga[1, 3*]

**1** IFW Dresden, Helmholtzstr. 20, 01069 Dresden, Germany
**2** Condensed Matter Physics and Material Sciences Department,
S. N. Bose National Centre for Basic Sciences, Kolkata, West Bengal-700106, India
**3** Würzburg-Dresden Cluster of Excellence *ct.qmat*, Germany

⋆ i.c.fulga@ifw-dresden.de

## Abstract

We show that the cubic compound PbBi$_2$ is a topological semimetal hosting a sixfold band touching point in close proximity to the Fermi level. Using angle-resolved photoemission spectroscopy, we map the band structure of the system, which is in good agreement with results from density functional theory. Further, by employing a low energy effective Hamiltonian valid close to the crossing point, we study the effect of a magnetic field on the sixfold fermion. The latter splits into a total of twenty Weyl cones for a Zeeman field oriented in the diagonal, (111) direction. Our results mark cubic PbBi$_2$ as an ideal candidate to study the transport properties of gapless topological systems beyond Dirac and Weyl semimetals.

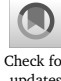

doi:10.21468/SciPostPhys.10.1.004

# 1 Introduction

Topological semimetals are materials whose band structure contains topologically protected intersections of energy bands. Well-known examples include Dirac and Weyl semimetals, that host point-like, cone-shaped band crossings [1–4]. They are governed by low energy effective Hamiltonians which take the same form as those describing their high energy physics namesakes, the elementary Dirac and Weyl fermions. Unlike in quantum field theory, however, Poincaré symmetry does not constrain the set of emergent quasiparticles allowed in condensed matter systems. As such, the low energy excitations of a crystal may come in flavors that go beyond those of the standard model of particle physics [5–16]. In terms of the band structure, these unconventional fermions arise at the intersections of multiple energy levels, such as 3, 4, 6, or even 8-fold points.

Beyond their appeal as a playground that harbors aspects of the fundamental theories of high energy physics, topological semimetals might also have potential technological applications. For the latter, one aims to exploit the unique electrical properties induced by the crossing points, such as the chiral anomaly [17], or the axial gravitational anomaly [18]. Since these effects involve mainly states close to the Fermi level, their observation imposes strict constraints on the energy at which band intersections should occur, thus reducing the range of viable material candidates. Indeed, despite the relatively large number of confirmed Weyl and Dirac systems [19–41], few show band crossings in close proximity to the Fermi level. For semimetals hosting unconventional fermions the list is smaller still, especially given that to date, few such materials have actually been synthesized and checked experimentally for the presence of nodal fermions [42–50].

In this work, we identify cubic $PtBi_2$ as an ideal candidate for transport studies on topological semimetals. We use a combination of angle-resolved photoemission spectroscopy (ARPES) and density functional theory (DFT) to map the band structure of the system. We show that cubic $PtBi_2$ hosts a sixfold band touching point – a triple Dirac point – in close proximity (10-20 meV) to the Femi level. To study the essential topological features of the crossing, we use a low-energy effective Hamiltonian which reproduces the ab-initio data. Upon including a weak Zeeman field to the model, we find that the sixfold point can split into a total of twenty type-II Weyl cones.

# 2 Band structure mapping

## 2.1 Theory

The DFT calculations where performed using the full 4-component relativistic mode of the Full Potential Local Orbital (FPLO) code [51] in version 15.02 with a $10 \times 10 \times 10$ k-mesh for the Brillouin zone (BZ) integration and the local density approximation (LDA) [52]. The space group is 205 ($Pa\bar{3}$) with lattice parameter $a_0 = 6.6954$ and Pt at Wyckoff position $(\frac{1}{2}, 0, 0)$ and Bi at $(0.12913, 0.12913, 0.12913)$.

Following the self consistent calculation a maximally projected Wannier function model [53, 54] containing all Bi $6p$- and Pt $5d$-orbitals was constructed. The corresponding core energy window reaches from $-5$ eV to $-3$ eV with a lower and upper Gausian fall-off of halfwidth 1 eV

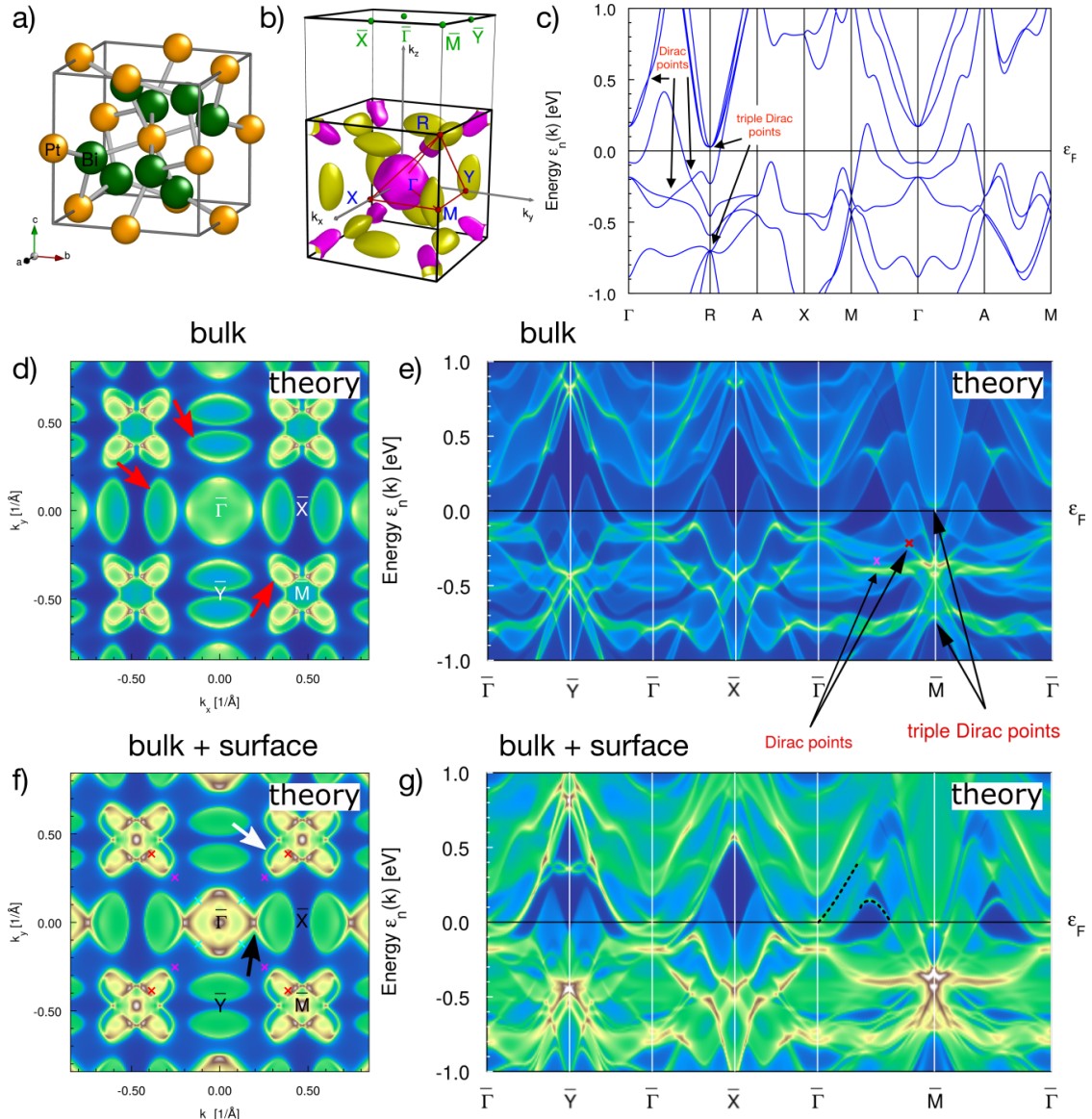

Figure 1: (color online) (a) the bulk unit cell, (b) the DFT 3D bulk Fermi surface with electron pockets (magenta) around the Γ- and R-points and hole pockets (yellow) on the line ΓM. (c) the DFT bulk band structure with arrows indicating the positions of Dirac and triple Dirac points. (d-g) DFT spectral densities in the extendend zone scheme at the Fermi level (d and f) and as energy-momentum cuts (e and g), (d) and (e) show $k_z$-integrated [(001)-direction] bulk projected bands, while (f) and (g) show the result for a BiPtBi-terminated [(001)-direction] semi infinite slab. The crosses show the position of the Dirac points along ΓR or $\overline{\Gamma M}$. Dashed lines in (g) highlight some important surface states dispersion. The red arrows in panel (d) mark features we refer to as "beans", whereas black and white arrows in (f) refer to "brackets" and "petals," respectively.

and 3 eV attached, respectively, ensuring a very good fit to the DFT band structure in an energy window from −6 eV to +5 eV with a maximum error around and below the Fermi level of 30meV.

This Wannier model was used to calculate the spectral density for an infinite bulk system by $k_z$-integrating the bulk band structure including an imaginary part of the band energies of 20meV [(001)-bulk projected bands (BPB)]. Furthermore, a semi-infinite slab (SIS) with a triple layer BiPtBi as termination was set up to calculate spectral densities of an idealized (001)-surface (simple Hamiltonian cutoff at the surface). The spectral densities were obtained via a Green's function recursion method [55, 56].

For the subsequent comparison with the photoemission intensity from the (001) surface of PtBi$_2$ we have calculated the spectral function corresponding to the bulk (BPB) and surface states (SIS) in a repeated zone scheme. Since the $k_z$-resolution is a strongly material-dependent quantity and thus the degree of surface contribution is not a priori known, we present both momentum- [Fig. 1(d), 1(f)] and energy-momentum [Fig. 1(e), 1(g)] intensity distributions along high symmetry directions by either including (SIS) or excluding (BPB) the surface states. Full integration of the bulk states along the (001) direction (BPB) will allow us a rough estimate of the $k_z$-resolution when comparing to experiment. Comparison of panels 1(d) and 1(e) with 1(f) and 1(g) respectively reveals the contribution of the surface states, as predicted by theory.

Fig. 1(b) suggests that there are three types of the Fermi surfaces in PtBi$_2$ and all of them are strongly three-dimensional. The structure near the $\overline{\Gamma}$-point is due to the projection of a large electron-like Fermi surface (FS) [Fig. 1(d) and 1(f)]. The ones in the corners of the BZ are the projections of eight electron-like Fermi surfaces united to the shape similar to a stellated octahedron centered at the R-points. Other twelve "beans" are the hole-like pockets, made by the crossings along ΓM and ΓA. They are responsible for the four larger filled ellipses close to the sides of the surface Brillouin zone and for the four smaller ellipses overlapping with the projections of the FS in the corners. Note the C$_2$ symmetry of the intensity pattern in Fig. 1(d), which stems from the symmetry reduction of the non-symmorphic cubic space group due to the surface termination/$k_z$-integration. For instance, filled ellipses are closer to each other along $\overline{\Gamma Y}$, and positioned farther apart along the $\overline{\Gamma X}$ direction. The projection with the center at the $\overline{M}$-point is clearly asymmetric too. The surface adds features which make the C$_2$-symmetry even more pronounced [Fig. 1(f)]. These are the "brackets" connecting the $\overline{\Gamma}$-centered FS-projection with ellipses along $\overline{\Gamma X}$ as well as the vertical arc-like connections of the "petals" near $\overline{M}$-point. The surface also adds bright and well localized intensity spots at $\overline{\Gamma}$- and $\overline{M}$-points.

## 2.2 Experiment

PtBi$_2$ single crystals were grown as described in Ref. [57]. Angle-resolved photoemission spectroscopy (ARPES) measurements were carried out at "1$^3$-ARPES" facility at BESSY at a temperature of 1 K within a wide range of photon energies (60 - 100 eV) from the cleaved surface of high-quality single crystals. The overall energy and momentum resolutions were set to ~8 meV and ~0.013 $^{-1}$, correspondingly.

In Fig. 2 we show the ARPES Fermi surface maps recorded using photons of different energies and compare them with the calculated momentum distribution including surface states (SIS). All maps shown in Fig. 2 and in particular the pattern within the first Brillouin zone (white dashed box) are in qualitative agreement with each other, with theory [Fig. 2(c)], as well as with a previous study focusing on the surface states of PtBi$_2$ [58]. All the FS projections discussed earlier are seen in the experimental maps having very similar shapes and intensity patterns, including the C$_2$-symmetry. Moreover, the surface related features, such as "brackets" and arc-like connections near the corners of the BZ ("petals") are clearly visible as well. The similarity of all experimental maps taken using different photon energies clearly implies a very low $k_z$-resolution. The differ-

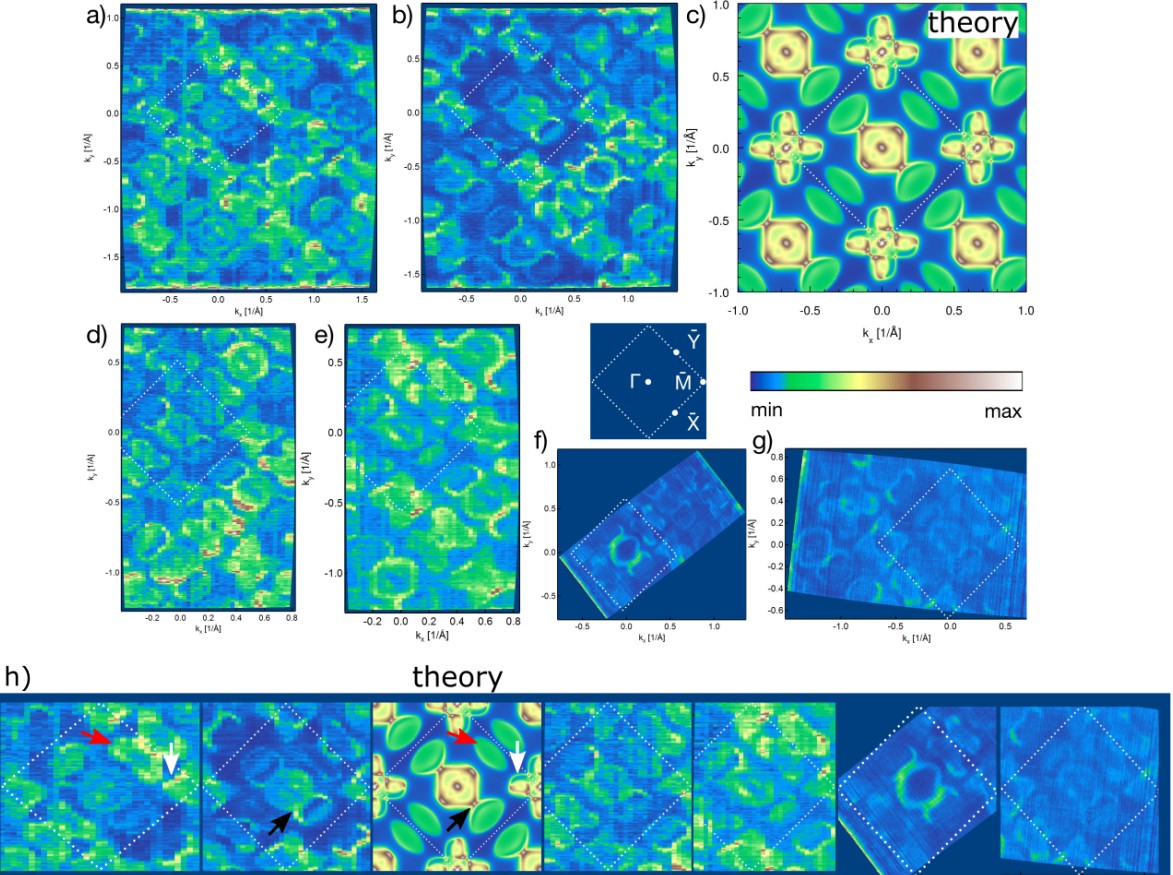

Figure 2: (color online) (a)-(b) and (d)-(g) ARPES Fermi surface maps at various photon energies: (a) 100 eV, (b, d) 80 eV, (e) 63 eV, (f) 70 eV, and (g) 60 eV. These maps are compared to (c) a suitably oriented DFT spectral density map at the Fermi energy including surface states (SIS). Panel (h) shows a side-by-side comparison of the first Brillouin zones shown in the other panels. The red, black, and white arrows indicate features referred to in the main text as "beans," "brackets," and "petals," respectively.

ences are mostly due to unequal intensity distribution rather than different peak positions of the spectral function. Therefore, one can consider any of the maps as representing the total spectral weight - fully integrated over $k_z$ bulk states plus surface states. Intensity variations are due to matrix elements highlighting a particular portion of the $k$-space.

Keeping this in mind, a closer inspection of the maps reveals certain discrepancies with theory. Ellipses are closer to each other in the experiment near $\overline{Y}$-points and even overlap or at least touch there. They also seem to be larger than their near-$\overline{X}$ counterparts. Another difference is the absence of the sharp intensity maxima due to surface states at the $\overline{\Gamma}$- and $\overline{M}$-points.

To get a deeper insight into the origin of the observed similarities and discrepancies we show in Fig. 3 the comparison of the experimental momentum-energy cuts with the corresponding calculations of the total intensity along the $\overline{\Gamma Y}$- and $\overline{\Gamma X}$-directions. In Figs. 3 and 4 we use binding energy for the vertical axis, which has a sign opposite to that of energy in Fig. 1. As in the case with the FS maps, the overall agreement is very good. All the well-pronounced features in the calculations can be tracked in the experimental data. Even the intensity distribution is captured

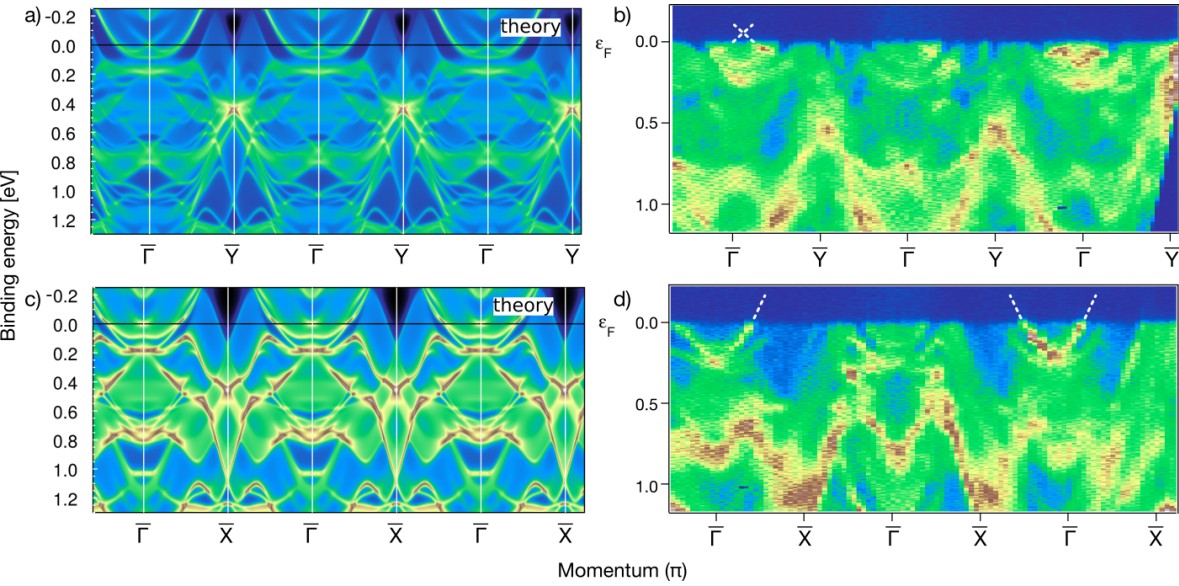

Figure 3: (color online) Comparison of the experimental (b and d) momentum-energy cuts with the DFT equivalent (a and c) in the setup including surface states (SIS) along the lines $\overline{\Gamma Y}$ (a and b) and $\overline{\Gamma X}$ (c and d). Dashed lines in (b and d) schematically show the position of some surface states dispersion above the Fermi level.

by the calculations correctly in most cases, but due to a big number of features their relative intensities are not always reproduced exactly. The most pronounced difference is the absence of the crossing of two dispersing features at the Fermi level at the $\overline{\Gamma}$-point. These features can be clearly identified as surface states, as follows from the comparison of Fig. 1(e) and Fig. 1(g). We noticed the corresponding absence of the bright spot in the Fermi surface maps earlier.

The data shown in Fig. 3(b) provides an explanation for this discrepancy. Indeed, there are two features close to the $\overline{\Gamma}$ point with the dispersion opposite to the electron-like parabola supporting the large $\overline{\Gamma}$-centered Fermi surface, but they cross the Fermi level before they reach the $\overline{\Gamma}$ point. This is best seen when considering the left-most $\overline{\Gamma}$ point in Fig. 3(b). To highlight these dispersions we have schematically extended them above the Fermi level using dashed lines. The corresponding set of features is seen also in all experimental maps in Fig. 2 inside the large $\overline{\Gamma}$-centered FS. Also, one has to keep in mind that the idealized surface termination in the SIS calculations may not accurately reproduce the experimentally realized energy dispersion or the ones corresponding to a self-consistent slab calculation including all surface relaxations. Nonetheless, we note that the other set of surface states, the one responsible for "brackets," is reproduced remarkably well: the linear dispersions starting approximately at 200 meV below the Fermi level at $\overline{\Gamma}$ are clearly seen in Fig. 3(d) for the left-most and righ-most BZs.

In Fig. 4 we show the comparison of theory with experiment for the $\overline{\Gamma M}$-high-symmetry direction. We have selected two representative cuts with different intensity distributions to highlight different parts of the spectra. Both measurements again show an overall agreement with the calculations. Also this time the most noticeable variations can be attributed to the surface states. The difference between the plots in Fig. 1(e) and 1(g) suggests that purely surface-originated features (except the already discussed ones at $\overline{\Gamma}$) are the parabolic dispersion with the top located roughly in between $\overline{\Gamma}$ and $\overline{M}$ at +150 meV and features dispersing between −100 and +500 meV and

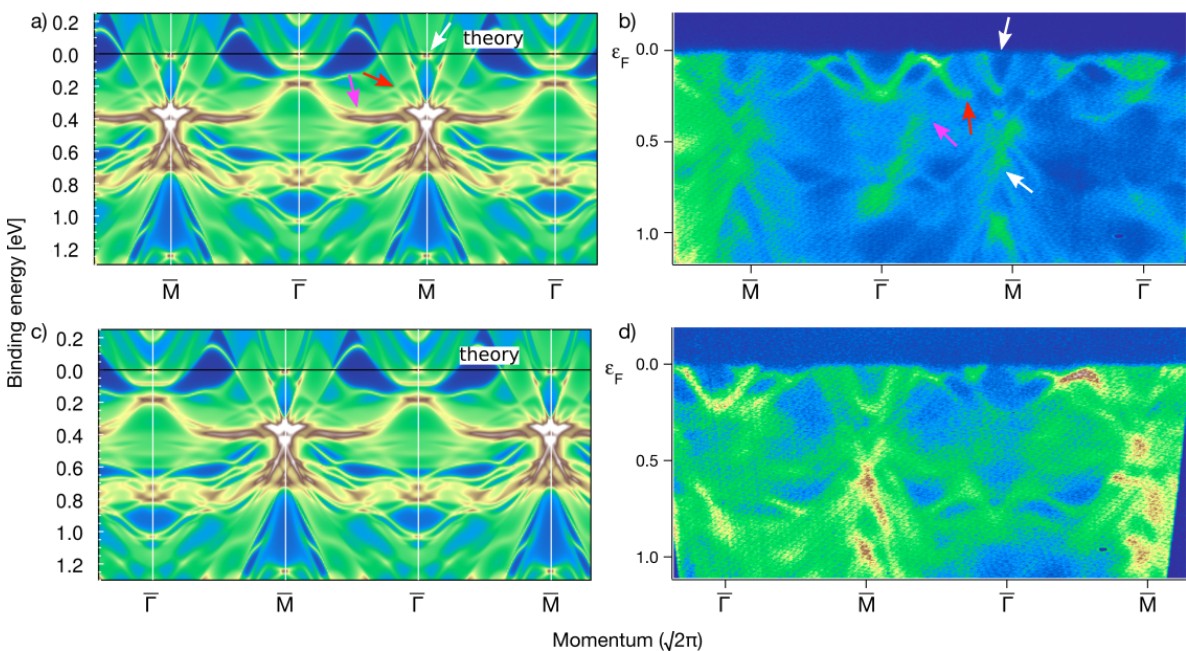

Figure 4: (color online) Comparison of two different experimental (b and d) momentum-energy cuts with the DFT equivalent (a and c) in the setup including surface states (SIS) along the line $\overline{\Gamma\mathrm{M}}$. The magenta and red arrows mark the position of the Dirac points and the white arrows the positions of the two triple Dirac points (at 700 and 15 meV below the Fermi level).

crossing near the Fermi level thus highlighting the $\overline{\mathrm{M}}$-point in the calculated map [dashed lines in Fig. 1(f)]. The experiment clearly shows that the mentioned parabolic dispersion becomes fully occupied. Indeed, the two "X"-like features are clearly seen between $\overline{\Gamma}$ and $\overline{\mathrm{M}}$ with the top of the above mentioned parabola residing at 90 meV below the Fermi level.

Taking into account that also the energetics of the $\overline{\Gamma}$-centered surface states is not fully accounted for by the calculations, it is not surprising that the surface states appear to be missing at the $\overline{\mathrm{M}}$-point - the bright spot is not there in the maps and we do not observe any corresponding crossing of features in Fig. 4(b) and 4(d). However, a weak feature at the Fermi level can be seen at the $\overline{\mathrm{M}}$-point [e.g. left $\overline{\mathrm{M}}$-point in Fig. 4(d)]. Taking into account that all surface states discussed above are the strongest features in terms of intensity, we attribute this one to the bulk states. Its shape corresponds to the bottom of the parabolic projection seen slightly above the Fermi level in Fig. 1(e) and thus to the triple Dirac point.

In spite of deviations in energy positions of some surface states the still observed overall agreement implies a remarkable correspondence between theory and experiment for the bulk states. Thus, the 3D Dirac points revealed by the calculations turn out to be experimentally realized. Using the positions of the Dirac points in the calculations one can trace their location in the experimental data. Because of the low $k_z$-resolution, we are not able to see the dispersions crossing in a single point directly, but the pattern of the features nearby allows us to identify the location of their projections [see arrows in Fig. 4(b)]. As mentioned above, the triple Dirac point is located at approximately 10-20 meV below the Fermi level, as indicated by one of the two white arrows in Fig. 4(b).

# 3 Low energy model

For the construction of the low energy model around the triple Dirac fermion point we used a fine resolution of the Wannier model (see Appendix).

The minimal model for a sixfold fermion of PtBi$_2$ (space group 205) can be found in Ref. [7]. The Hamiltonian is quadratic in momentum, $\mathbf{k} = (k_x, k_y, k_z)$, always measured relative to the band crossing point:

$$
H_{205} = \begin{pmatrix}
f_1(\mathbf{k}) & ae^{i\phi_a}k_y k_z & ae^{-i\phi_a}k_x k_z & 0 & be^{i\phi_b}k_y k_z & -be^{i\phi_b}k_x k_z \\
ae^{-i\phi_a}k_y k_z & f_2(\mathbf{k}) & ae^{i\phi_a}k_x k_y & -be^{i\phi_b}k_y k_z & 0 & be^{i\phi_b}k_x k_y \\
ae^{i\phi_a}k_x k_z & ae^{-i\phi_a}k_x k_y & f_3(\mathbf{k}) & be^{i\phi_b}k_x k_z & -be^{i\phi_b}k_x k_y & 0 \\
0 & -be^{-i\phi_b}k_y k_z & be^{i\phi_b}k_x k_z & f_1(\mathbf{k}) & ae^{-i\phi_a}k_y k_z & ae^{i\phi_a}k_x k_z \\
be^{-i\phi_b}k_y k_z & 0 & -be^{-i\phi_b}k_x k_y & ae^{i\phi_a}k_y k_z & f_2(\mathbf{k}) & ae^{-i\phi_a}k_x k_y \\
-be^{-i\phi_b}k_x k_z & be^{-i\phi_b}k_x k_y & 0 & ae^{-i\phi_a}k_x k_z & ae^{i\phi_a}k_x k_y & f_3(\mathbf{k})
\end{pmatrix}, \quad (1)
$$

where

$$
\begin{aligned}
f_1(\mathbf{k}) &= E_0 + E_1 k_x^2 + E_2 k_y^2 + E_3 k_z^2 \\
f_2(\mathbf{k}) &= E_0 + E_1 k_z^2 + E_2 k_x^2 + E_3 k_y^2 \\
f_3(\mathbf{k}) &= E_0 + E_1 k_y^2 + E_2 k_z^2 + E_3 k_x^2 .
\end{aligned} \quad (2)
$$

The symmetries which are relevant for the sixfold fermion are: a 3-fold screw in the (111) direction, denoted $S_3$, a 2-fold screw in the $x$ direction $[S_2]$, inversion $[I]$, and time-reversal symmetry (TRS, $T$). The representation of these operators is

$$
\begin{aligned}
S_3 &= \begin{pmatrix} 0 & 0 & 1 \\ 1 & 0 & 0 \\ 0 & 1 & 0 \end{pmatrix} \otimes \sigma_0 \\
S_2 &= \begin{pmatrix} -1 & 0 & 0 \\ 0 & -1 & 0 \\ 0 & 0 & 1 \end{pmatrix} \otimes \sigma_0 \\
I &= \mathbf{1}_{3\times3} \otimes \sigma_0 \\
T &= i\mathbf{1}_{3\times3} \otimes \sigma_y K ,
\end{aligned} \quad (3)
$$

where $K$ denotes complex conjugation, $\sigma$ are Pauli matrices in spin space, the remaining $3 \times 3$ matrices act in orbital space, and $\mathbf{1}_{3\times3}$ is the identity matrix.

Using ab-initio data on a momentum grid around the 6-fold fermion, we extracted the parameters $E_{0,1,2,3}$, $a$, $\phi_a$, and $b$ of the Hamiltonian Eq. (1) (see Appendix). Note that $\phi_b$ can be gauged away by the basis transformation $U = \text{diag}(1, 1, 1, e^{i\phi_b}, e^{i\phi_b}, e^{i\phi_b})$, and as such it will not affect the spectrum of the low-energy model. For this reason we set $\phi_b = 0$ throughout the following.

The magnetic field (here considered to be a Zeeman field) has two main effects. First, it breaks TRS, such that different spin states have in general different energies. Second, it breaks some of the lattice symmetries, depending on its orientation. On the level of the low-energy Hamiltonian, we model a Zeeman field as

$$
H_z = H_z^{\text{orb}} \otimes (B_x \sigma_x + B_y \sigma_y + B_z \sigma_z), \quad (4)
$$

which is separated into an orbital part that acts on the ortibal degrees of freedom, $H_z^{\text{orb}}$, and a spin part which is taken to be the usual Zeeman term. For the (111) direction, $B_x = B_y = B_z \equiv B$, and $H_z^{\text{orb}}$ is obtained as the most general $3 \times 3$ matrix which respects the 3-fold screw symmetry $[H_z, S_3] = 0$ [see Eq. (3)]. Following Ref. [7] we obtain

$$
H_z^{\text{orb}} = U\text{diag}(g_1 + g_0, g_2 + g_0, -g_1 - g_2 + g_0)U^\dagger, \quad U = \frac{1}{\sqrt{3}}\begin{pmatrix} 1 & e^{-\frac{2i\pi}{3}} & e^{\frac{2i\pi}{3}} \\ 1 & e^{\frac{2i\pi}{3}} & e^{-\frac{2i\pi}{3}} \\ 1 & 1 & 1 \end{pmatrix}. \quad (5)
$$

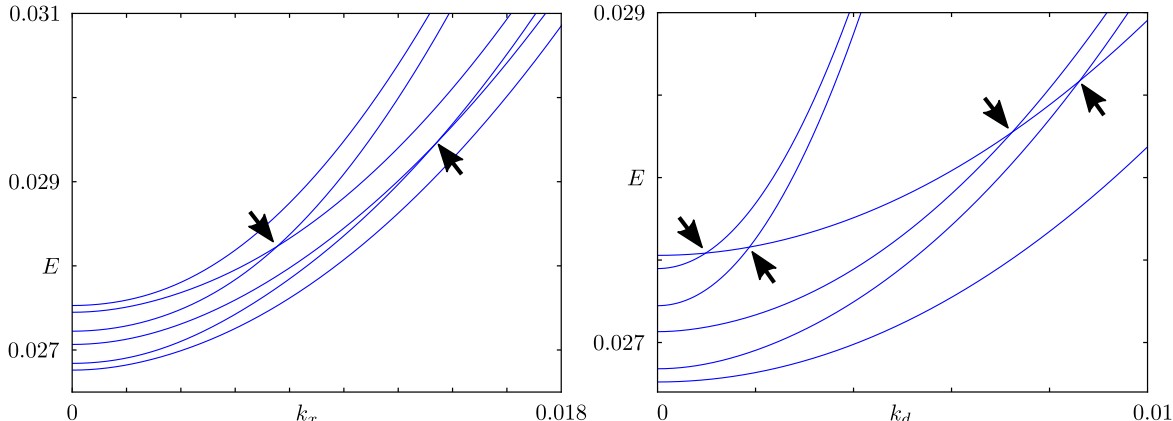

Figure 5: Band structure in the presence of a (111) Zeeman field ($B = 3 \cdot 10^{-4}$). The left panel shows the $k_x$ direction for $k_y = k_z = 0$, and the bottom panel shows the $k_x = k_y = k_z \equiv k_d$ direction. Weyl cones are marked by black arrows.

As before, we determine the $g$ factors by comparing the model to the ab-initio data, with details shown in the Appendix.

When the Zeeman field is turned on, the 6-fold crossing splits into a total of 20 Weyl cones. There are 8 Weyl nodes on the (111) axis, and 4 on each of the three principal directions (totaling 12), which are related to each other by the 3-fold screw symmetry. Because the Zeeman term preserves inversion symmetry, the band crossings are $k \to -k$ symmetric, with Weyl cones at opposite $k$ having opposite charge.

In the following, we fix $B = 3 \cdot 10^{-4}$ and show the Weyl node positions and charges. We find that all 20 Weyl cones are type-II, or over-tilted. Figure 5 shows the spectrum as a function of $k_x$ for $k_x = k_y = 0$ (left panel), as well as along $k_x = k_y = k_z \equiv k_d$ (right panel). In both cases, Weyl points are marked with arrows.

To show that there are no other band crossing points except for the ones mentioned above, we have performed a 3D scan in momentum space, as shown in Fig. 6. In this figure we also indicate the monopole charge of each node, obtained by numerically integrating the Berry curvature over a closed surface surrounding each Weyl cone. In the Supplemental Material, we have included a code which implements the low energy model and numerically determines the monopole charge of each band crossing point, thus confirming that they are Weyl nodes.

## 4 Conclusion

Crossings of energy levels such as Dirac points, Weyl points, and their multi-fold generalizations occur in a variety of band structures, but few materials host them in the vicinity of the Fermi level. We have shown that cubic PtBi$_2$ is one such material. We have performed angle-resolved photoemission spectroscopy to map its band structure, finding good agreement with density functional theory calculations. Our results show that PtBi$_2$ hosts a symmetry protected crossing of six energy bands, 50 meV above $E_F$ in DFT and 10-20 meV below $E_F$ according to ARPES.

Starting from the ab-initio results, we have constructed a low-energy effective model for the sixfold fermion, which allowed us to study the effect of a Zeeman field on the band crossing. Remarkably, for a magnetic field pointing in the (111) direction, the band crossing splits into a total of twenty Weyl cones, marking PtBi$_2$ as an ideal candidate for magnetotransport studies. We hope that our work will motivate further research in this direction, including measurements of the chiral anomaly and studies of non-linear transport effects generated by Berry curvature dipoles [59–61].

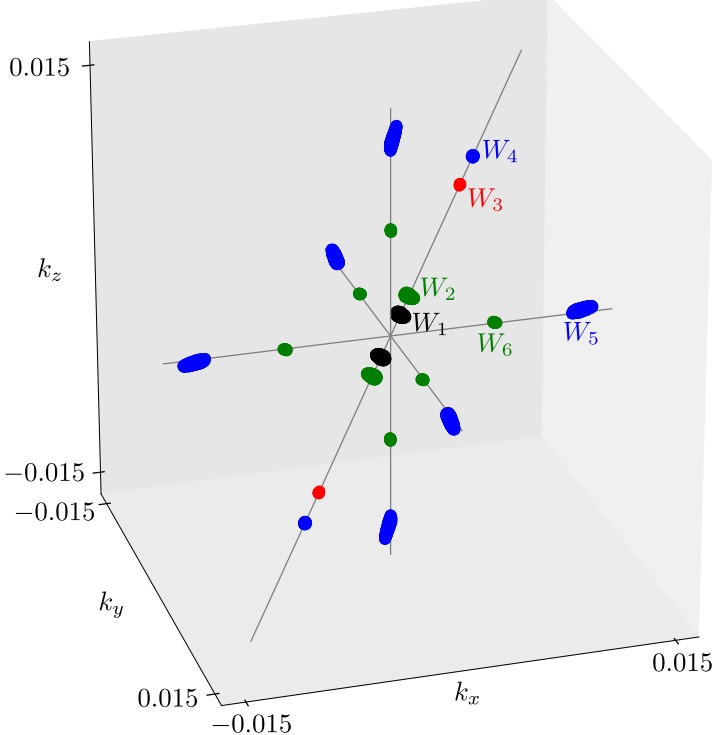

Figure 6: Position of Weyl nodes in the presence of a (111) Zeeman field ($B = 3 \cdot 10^{-4}$). Gray lines indicate the three principal $k_{x,y,z}$ directions, as well as the diagonal $k_x = k_y = k_z$ direction. Ordering the eigenvalues in increasing energy from 1 to 6, colored patches indicate crossings between levels 2 and 3 (blue), 3 and 4 (red), 4 and 5 (green), as well as between levels 5 and 6 (black). Each crossing is determined by the set of momenta for which the energy difference of neighboring bands is less than $10^{-5}$. Six inequivalent Weyl cones are labeled $W_j$, $j = 1, \ldots, 6$. All other crossing points can be inferred by applying inversion or 3-fold rotation. Their monopole charge is $Q = +1$ for $W_1$, $W_4$, and $W_6$, whereas it takes the value $Q = -1$ for $W_2$, $W_3$, and $W_5$.

# Acknowledgements

We thank Ulrike Nitzsche for technical assistance.

**Funding information**    This work is supported by the Deutsche Forschungsgemeinschaft (DFG, German Research Foundation) through the Würzburg-Dresden Cluster of Excellence on Complexity and Topology in Quantum Matter – *ct.qmat* (EXC 2147, project-id 390858490), and through the grants BO1912/7-1 and SFB1143. S. T. acknowledges the financial support by DST, India through the INSPIRE-Faculty program (Grant No. IFA14 PH-86), as well as the financial support given by SNBNCBS through the Faculty Seed Grants program. B. B. and S. A. acknowledge financial support by the DFG-RSF joint project id:405940956. S. A. acknowledges the financial support by DFG AS 523/4-1. S. B., Y. K., and B. B. acknowledge BMBF through the UKRATOP project.

# A   Model parameters from ab-initio data

Here we describe the method used to extract the parameters of the low-energy model Eq. (1), given the band structure generated in DFT.

The simplest to estimate is $E_0$, the energy of the band crossing point, obtained by setting $\mathbf{k} = 0$. Setting $k_y = k_z = 0$ instead, the Hamiltonian is diagonal, allowing us to obtain $E_{1,2,3}$, which are just the curvatures of the three parabolic bands (each band is doubly degenerate because of inversion and TRS). Note that the ab-initio data is on a discrete grid, so to get the curvatures we used the $k_x$ points closest to the 6-fold crossing ($k_x = 0.02$ in units of $2\pi/a$). This is necessarily an approximation, however, because Eq. (1) is only valid infinitesimally away from $\mathbf{k} = 0$, so that for any finite distance away there will be a (small) contribution from higher-order momentum terms. We find $E_0 \simeq 0.027$, $E_1 \simeq 12.192$, $E_2 \simeq 13.855$, and $E_3 \simeq 20.927$, where we have chosen $E_1 < E_2 < E_3$ without loss of generality. All parabolas have positive curvature.

To extract the remaining parameters, we use the eigenvalues of the system along the diagonal momentum direction $k_d \equiv k_x = k_y = k_z$, which now only depend on $b$, $a\cos(\phi_a)$, and $a\sin(\phi_a)$. As before, each level $\varepsilon$ is 2-fold degenerate

$$
\begin{aligned}
\varepsilon_1 &= E_0 + k_d^2 \left[ E_1 + E_2 + E_3 + 2a\cos(\phi_a) \right] \\
\varepsilon_{2,3} &= E_0 + k_d^2 \left[ E_1 + E_2 + E_3 - a\cos(\phi_a) \mp \sqrt{3\left[ b^2 + a^2\sin(\phi_a)^2 \right]} \right].
\end{aligned}
\tag{6}
$$

Again, comparing to the levels closest to the crossing, at $k_d = 0.02$, allows to extract the Hamiltonian parameters. Using the $E_{0,1,2,3}$ found before, this is a system of 3 equations with 3 unknowns, but it does not have an exact solution, again because of higher order momentum terms which are nonzero away from $\mathbf{k} = 0$. For this reason, we use instead a numerical minimization routine to find the best $a$, $b$, and $\phi_a$. The routine minimizes $\sum_{j=1,2,3} |\varepsilon_j^{k\cdot p} - \varepsilon_j^{\text{DFT}}|$, where $\varepsilon_j^{\text{DFT}}$ are the energies obtained by DFT, finding $a \simeq -24.666$, $b \simeq 14.365$, and $\phi_a \simeq -0.846$. The largest relative error introduced by this process is $\max_{j=1,2,3} |\varepsilon_j - \varepsilon_j^{\text{DFT}}| / \varepsilon_j^{\text{DFT}} = 0.01529$, so less than 1.6%. Finally, note that one can always change $\phi_a \to \phi_a + \pi$ and $a \to -a$ without changing the spectrum, simply because the two always appear as $ae^{\pm i\phi_a}$ in the Hamiltonian.

To estimate the $g$ factors entering the Zeeman term (4), we use a similar numerical minimization algorithm. To this end, we added to the non-magnetic ab-initio Hamiltonian a term $\langle \vec{\Sigma} \rangle \cdot \vec{B}$ where $\langle \vec{\Sigma} \rangle$ is the matrix of the spin operator in the DFT basis, which allows to have a Zeeman field that acts correctly on the individual orbitals. We used a diagonal $B = 0.01$ and extracted the eigenvalues at $\mathbf{k} = 0$:

$$
\begin{aligned}
\varepsilon_{1,2} &= E_0 \mp \sqrt{3}B|g0 + g1| \\
\varepsilon_{3,4} &= E_0 \mp \sqrt{3}B|g0 + g2| \\
\varepsilon_{5,6} &= E_0 \mp \sqrt{3}B|-g0 + g1 + g2|.
\end{aligned}
\tag{7}
$$

With the $E_0$ found before, we obtain $g_0 \simeq 0.498$, $g_1 \simeq 0.237$, and $g_2 \simeq 0.076$. The relative errors of energy levels obtained through this numerical minimization (as compared to DFT) are 5.13%, 2.82%, 1.68%, 1.29%, 0.63%, and 0.18%. To get a better estimate, one can also include a linear in $B$ total energy shift of the bands, by adding a term $Bg_{\text{shift}}\mathbf{1}_{6\times 6}$ and fitting $g_{\text{shift}}$ together with the other $g$ factors. Doing this results in $g_0 \simeq 0.391$, $g_1 \simeq 0.348$, $g_2 \simeq 0.194$, and $g_{\text{shift}} \simeq 0.047$. Notice that $g_{\text{shift}}$ is significantly smaller than the other terms. The relative errors in this case are also smaller: 1.09%, 0.97%, 0.55%, 0.017%, and the remaining two smaller than $10^{-6}$%. In the plots shown in Figs. 5 and 6, we have included this overall energy shift.

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
