# Peer review of "Sixfold fermion near the Fermi level in cubic PtBi2"

_SciPost Physics_

## Round 1 · Referee Report · Anonymous (Referee 1) · 2020-7-24

Report

Authors study electronic properties of PtBi2 using DFT calculations. They find that this material has sixfold band touching (sixfold fermion). The manuscript is well written and clearly argues the theoretical and experimental case. The only room for improvement would be figures:

1) I would suggest to use different color scheme for calculations and experimental data.

2) The FS data in Fig. 2 is plotted over many Brillouin zones (BZ) which makes it very unclear. If authors want to show multi zone plot I would suggest using just a single panel, then plotting only single BZ for each photon energy, so reader can see better the differences.

3) Fig 3 and 4 are key to the claims of the paper. The crosses obscure the data and it is not clear that the bands form the Dirac points, let alone six fold ones. Perhaps adding a panel or figure that shows more clearly the Dirac points and all bands could resolve the issue. The authors could also try second derrivative technique to better visualize the bands.

  • validity: -
  • significance: -
  • originality: -
  • clarity: -
  • formatting: -
  • grammar: -

Author:  Ion Cosma Fulga  on 2020-11-03  [id 1028]

(in reply to Report 1 on 2020-07-24)

Please see the attached pdf for our reply.

Attachment:

reply.pdf

---

## Round 1 · Referee Report · Anonymous (Referee 2) · 2020-9-4

Report

The authors report the experimental realization of a high order nodal point semimetal PtBi2. The experimental data and theory calculations are solid, and it deserves publication. I have a couple of objections that should be addressed.

  1. Could the author use arrows also in Fig. (2) to be able to follow the explanation of the text ?
  2. They claim under a Zeeman field the bands split in 20 Weyl points. They should explain why they are all Weyl nodes.
  • validity: good
  • significance: good
  • originality: good
  • clarity: high
  • formatting: excellent
  • grammar: excellent

Author:  Ion Cosma Fulga  on 2020-11-03  [id 1029]

(in reply to Report 2 on 2020-09-04)

Please see attached pdf for our reply.

Attachment:

reply_bZcCq2s.pdf

Anonymous on 2020-11-23  [id 1056]

(in reply to Ion Cosma Fulga on 2020-11-03 [id 1029])

The authors have addressed my comments and therefor I recommend the manuscript for publication.

---

## Editorial Decision

resubmitted